# UNCONDITIONAL SYNTHESIS OF COMPLEX SCENES USING A SEMANTIC BOTTLENECK

## ABSTRACT

Coupling the high-fidelity generation capabilities of label-conditional image synthesis methods with the flexibility of unconditional generative models, we propose a semantic bottleneck GAN model for unconditional synthesis of complex scenes. We assume pixel-wise segmentation labels are available during training and use them to learn the scene structure through an unconditional progressive segmentation generation network. During inference, our model first synthesizes a realistic segmentation layout from scratch, then synthesizes a realistic scene conditioned on that layout through a conditional segmentation-to-image synthesis network. When trained end-to-end, the resulting model outperforms state-of-the-art generative models in unsupervised image synthesis on two challenging domains in terms of the Fréchet Inception Distance and perceptual evaluations. Moreover, we demonstrate that the end-to-end training significantly improves the segmentation-to-image synthesis sub-network, which results in superior performance over the state-of-the-art when conditioning on real segmentation layouts.

## 1 INTRODUCTION

Significant strides have been made on generative models for image synthesis, with a variety of methods based on Generative Adversarial Networks (GANs) (Goodfellow et al., 2014) achieving state-of-the-art performance. At lower resolutions or in specialized domains, GAN-based methods are able to synthesize samples which are near-indistinguishable from real samples (Brock et al., 2019). However, generating complex, high-resolution scenes from scratch remains a challenging problem, as shown in Figure 1-(a) and (b). As image resolution and complexity increase, the coherence of synthesized images decreases — samples lack consistent local or global structures.

Stochastic decoder-based models, such as conditional GANs, were recently proposed to alleviate some of these issues. In particular, both Pix2PixHD (Wang et al., 2018) and SPADE (Park et al., 2019) are able to synthesize high-quality scenes using a strong conditioning mechanism based on semantic segmentation labels during the scene generation process. Global structure encoded in the segmentation layout of the scene is what allows these models to focus primarily on generating convincing local content consistent with that structure.

A key practical drawback of such conditional models is that they require full segmentation layouts as input. Thus, unlike unconditional generative approaches which synthesize images from randomly sampled noise, these models are limited to generating images from a set of scenes that is prescribed in advance, typically either through segmentation labels from an existing dataset, or scenes that are hand-crafted by experts.

**Contributions** To overcome these limitations, we propose a new model, the Semantic Bottleneck GAN (SB-GAN), which couples high-fidelity generation capabilities of label-conditional models with the flexibility of unconditional image generation. This in turn enables our model to synthesize an unlimited number of novel complex scenes, while still maintaining high-fidelity output characteristic of image-conditional models.

Our SB-GAN first unconditionally generates a pixel-wise semantic label map of a scene (i.e. for each spatial location it outputs a class label), and then generates a realistic scene image by conditioning on that semantic map, Figure 1-(d). By factorizing the task into these two steps, we are able

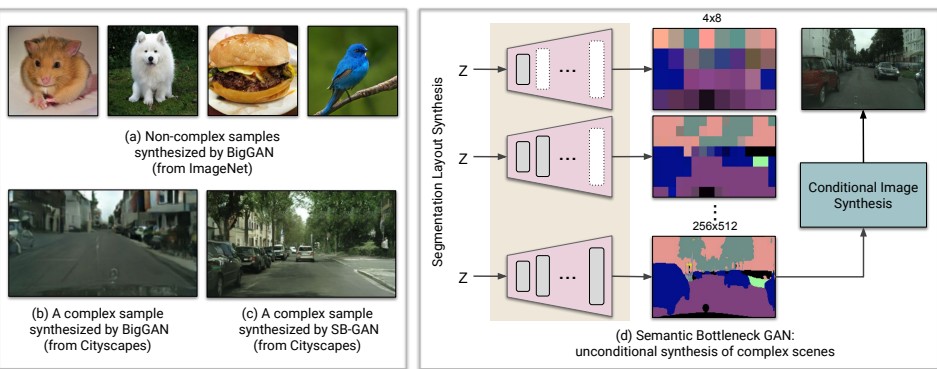

Figure 1: (a) Examples of non-complex images from ImageNet synthesized by the state-of-the-art BigGAN model (Brock et al., 2019). Although these samples look decent, the complex scenes synthesized by BigGAN (e.g., from the Cityscapes dataset) are blurry and defective in local structure (e.g., cars are blended together) (b). Zoom in for more detail. (c) A complex scene synthesized by our model respects both local and global structural integrity of the scene. (d) Schematic of our unconditional Semantic Bottleneck GAN. We progressively train the adversarial segmentation synthesis network to generate realistic segmentation maps from scratch, then synthesize a photo-realistic image using a conditional image synthesis network. End-to-end coupling of these two components results in state-of-the-art unconditional synthesis of complex scenes.

to separately tackle the problems of producing convincing segmentation layouts (i.e. a useful global structure) and filling these layouts with convincing appearances (i.e. local structure). When trained end-to-end, the model yields samples which have a coherent global structure as well as fine local details, e.g., Figure 1-(c). Empirical evaluation shows that our Semantic Bottleneck GAN achieves a new state-of-the-art on two complex datasets with relatively small number of training images, Cityscapes and ADE-Indoor, as measured both by the Fréchet Inception Distance (FID) and by perceptual evaluations. Additionally, we observe that the conditional segmentation-to-image synthesis component of our SB-GAN jointly trained with segmentation layout synthesis significantly improves the state-of-the-art semantic image synthesis network (Park et al., 2019), resulting in higher-quality outputs when conditioning on ground truth segmentation layouts.

**Key Challenges**   While both unconditional generation and image-to-image translation are well-explored learning problems, fully unconditional generation of the segmentation maps is a notoriously hard task: (i) Semantic categories do not respect any ordering relationships and the network is therefore required to capture the intricate relationship between segmentation classes, their shapes, and their spatial dependencies. (ii) As opposed to RGB values, semantic categories are discrete, hence non-differentiable which poses a challenge for end-to-end training (Sec. 3.2) (iii) Naively combining state-of-the-art unconditional generation and image-to-image translation models leads to poor performance. However, by carefully designing an additional discriminator component and a corresponding training protocol, we not only manage to improve the performance of the end-to-end model, but also the performance of each component separately (Sec. 3.3).

We emphasize that despite these challenges our approach scales to $256 \times 256$ resolution and 95 semantic categories, whereas existing state-of-the-art GAN models directly generating RGB images at that resolution already suffer from considerable instability (Sec. 4).

## 2   RELATED WORK

**Generative Adversarial Networks (GANs)**   GANs (Goodfellow et al., 2014) are a powerful class of generative models successfully applied to various image synthesis tasks such as image style transfer (Isola et al., 2017; Zhu et al., 2017), unsupervised representation learning (Chen et al., 2016; Pathak et al., 2016; Radford et al., 2016), image super-resolution (Ledig et al., 2017; Dong et al., 2016), and text-to-image synthesis (Zhang et al., 2017; Xu et al., 2018; Qiao et al., 2019b). Training GANs is notoriously hard and recent efforts focused on improving neural architectures (Wang & Gupta, 2016; Karras et al., 2017; Zhang et al., 2019; Chen et al., 2019a), loss functions (Arjovsky et al., 2017), regularization (Gulrajani et al., 2017; Miyato et al., 2018), large-scale training (Brock

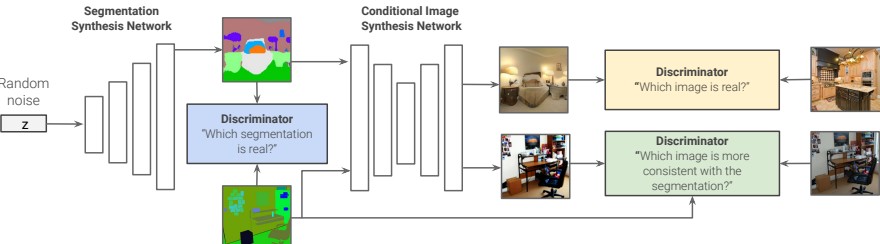

Figure 2: Schematic of Semantic Bottleneck GAN. Starting from random noise, we synthesize a segmentation layout and use a discriminator to bias the segmentation synthesis network towards realistic looking segmentation layouts. The generated layout is then provided as input to a conditional image synthesis network to synthesize the final image. A second discriminator is used to bias the conditional image synthesis network towards realistic images paired with real segmentation layouts. Finally, a third unconditional discriminator is used to bias the conditional image synthesis network towards generating images that match the real image distribution.

et al., 2019), self-supervision (Chen et al., 2019b), and sampling (Brock et al., 2019; Azadi et al., 2019a). Improving the performance of GANs by disentangling structure and style has been studied by Wang & Gupta (2016) where structure is represented by a surface normal map and style is the texture mapped onto the structure. Another compelling approach which enables generation of high-resolution images is based on progressive training: a model is trained to first synthesize lower-resolution images (e.g. $8 \times 8$), then the resolution is gradually increased until the desired resolution is achieved (Karras et al., 2017). Recently, Brock et al. (2019) showed that GANs significantly benefit from large-scale training, both in terms of model size and batch size. We note that these models are able to synthesize high-quality images in settings where objects are very prominent and centrally placed or follow some well-defined structure, as the corresponding distribution is easier to capture. In contrast, when the scenes are more complex and the amount of data is limited, the task becomes extremely challenging for these state-of-the-art models. We aim to improve the performance in the context of complex scenes and a small number of training examples by disentangling the image generation problem into learning the structure represented by semantic layouts and filling in the RGB details using a semantic image synthesis model. A similar idea was proposed by a concurrent work (Volokitin et al., 2020) with substantial differences in the model and results.

**GANs on discrete domains** GANs for discrete domains have been investigated in several works (Yu et al., 2017b; Lin et al., 2017; Bojchevski et al., 2018; Lu et al., 2018). Training in this domain is even more challenging as the samples from discrete distributions are not differentiable with respect to the network parameters. This problem can be somewhat alleviated by using the Gumbel-softmax distribution, which is a continuous approximation to a multinomial distribution parameterized in terms of the softmax function (Kusner & Hernández-Lobato, 2016). We will show how to apply a similar principle to learn the distribution of discrete segmentation masks.

**Conditional image synthesis** In conditional image synthesis one aims to generate images by conditioning on an input which can be provided in the form of an image (Isola et al., 2017; Zhu et al., 2017; Azadi et al., 2018; 2019b; Liu et al., 2017), a text phrase (Reed et al., 2016; Zhang et al., 2017; Qiao et al., 2019a; Ashual & Wolf, 2019; Hong et al., 2018), a scene graph (Johnson et al., 2018; Ashual & Wolf, 2019), a class label, or a semantic layout (Odena et al., 2017; Chen & Koltun, 2017; Wang et al., 2018; Park et al., 2019). These conditional GAN methods learn a mapping that translates samples from the source distribution into samples from the target domain.

The text-to-image synthesis models proposed in (Hong et al., 2018; Li et al., 2019) decompose the synthesis task into multiple steps. As illustrated in the Appendix, given the text description, a semantic layout is constructed by generating object bounding boxes and refining each box by estimating object shapes. Then, an image is synthesized conditionally on the generated semantic layout from the first step. Our work shares the same high-level idea of decomposing the image generation problem into the semantic layout synthesis and the conditional semantic-layout-to-image synthesis. However, we note that the above approaches, as opposed to ours, are conditional and require supervision in the form of textual descriptions. Secondly, they are sequential in nature and synthesize masks of a few different objects (e.g. person, elephant), but not a fully fine-grained semantic map (e.g. missing sky, grass, etc.). In stark contrast, our approach unconditionally synthesizes the *full*

*semantic layout* of the entire scene from a noise input in an end-to-end network design. Due to the above distinctions, their segmentation synthesis models differ significantly from ours in terms of architecture and design as shown in Figure 6 in the Appendix.

## 3 SEMANTIC BOTTLENECK GAN (SB-GAN)

We propose an unconditional Semantic Bottleneck GAN architecture to learn the distribution of complex scenes. To tackle the problems of learning both the global layout and the local structure, we divide this synthesis problem into two parts: an unconditional segmentation map synthesis network and a conditional segmentation-to-image synthesis model. Our first network is designed to coarsely learn the scene distribution by synthesizing semantic layouts. It generates per-pixel semantic categories following the progressive GAN model architecture (ProGAN) (Karras et al., 2017). This fully unconditional generation of the segmentation maps is novel, very challenging, and a careful design is crucial, as described in Section 3.1. The second network populates the synthesized semantic layouts with texture by predicting RGB pixel values using Spatially-Adaptive Normalization (SPADE), following the architecture of the state-of-the-art semantic synthesis network in (Park et al., 2019). We assume the ground truth segmentation masks are available for all or part of the target scene dataset. In the following sections, we will first discuss our semantic bottleneck synthesis pipeline and summarize the SPADE network for image synthesis. We will then couple these two networks in an end-to-end design which we refer to as Semantic Bottleneck GAN (SB-GAN).

### 3.1 SEMANTIC BOTTLENECK SYNTHESIS

Our goal is to learn a (coarse) estimate of the scene distribution from samples corresponding to real segmentation maps with $K$ semantic categories. Starting from random noise, we generate a tensor $Y \in [\![1, K]\!]^{N \times 1 \times H \times W}$ which represents a per-pixel segmentation class, with $H$ and $W$ indicating the height and width, respectively, of the generated map and $N$ the batch size. In practice, we progressively train from a low to a high resolution using the ProGAN architecture (Karras et al., 2017) coupled with the Improved WGAN loss function (Gulrajani et al., 2017) on the ground truth discrete-valued segmentation maps, illustrated in Figure 1-(d). Similar to ProGAN, to increase the spatial resolution of the generated segmentation maps during training, we incrementally add layers to the generator and the discriminator. In contrast to ProGAN, in which the generator outputs continuous RGB values, we predict per-pixel discrete semantic class labels. This task is extremely challenging as it requires the network to capture the intricate relationship between segmentation classes and their spatial dependencies. To this end, we apply the Gumbel-softmax trick (Jang et al., 2017; Maddison et al., 2016) coupled with a straight-through estimator (Jang et al., 2017), described in detail below.

We synthesize segmentation layouts by first generating per-pixel probability scores of belonging to each of the $K$ semantic classes and then sampling a semantic class per pixel. The per-pixel probability scores are computed by applying a softmax function to the last layer of the generator (i.e. logits) which results in probability maps $P^{ij} \in [0, 1]^K$, with $\sum_{k=1}^{K} P_k^{ij} = 1$ for each spatial location $(i, j) \in [\![1, H]\!] \times [\![1, W]\!]$. To sample a semantic class from this multinomial distribution, we would ideally apply the following well-known procedure at each spatial location: (1) sample $k$ i.i.d. samples, $G_k$, from the standard Gumbel distribution, (2) add these samples to each logit, and (3) take the index of the maximal value. This reparametrization indeed allows for an efficient forward-pass, but is not differentiable. Nevertheless, the `max` can be replaced with the `softmax` function and the quality of the approximation can be controlled by varying the *temperature hyperparameter* $\tau$ — the smaller the $\tau$, the closer the approximation is to the categorical distribution (Jang et al., 2017):

$$S_k^{ij} = \frac{\exp\{(\log P_k^{ij} + G_k)/\tau\}}{\sum_{i=1}^{K} \exp\{(\log P_i^{ij} + G_i)/\tau\}}. \tag{1}$$

Similar to the real samples, the synthesized samples fed to the GAN discriminator should still contain *discrete* category labels. As a result, for the forward pass, we compute $\arg\max_k S_k$, while for the backward pass, we use the soft predicted scores $S_k$ directly, a strategy known as straight-through estimation (Jang et al., 2017).

## 3.2 SEMANTIC IMAGE SYNTHESIS

Our second sub-network converts the synthesized semantic layouts into photo-realistic images using spatially-adaptive normalization (Park et al., 2019). The segmentation masks are employed to spread the semantic information throughout the generator by modulating the activations with a spatially adaptive learned transformation. We follow the same generator and discriminator architectures and loss functions used in (Park et al., 2019), where the generator contains a series of SPADE residual blocks with upsampling layers. The loss functions to train SPADE are summarized as:

$$
\begin{aligned}
L_{D_{\text{SPD}}} &= -\mathbb{E}_{y,x}[\min(0, -1 + D_{\text{SPD}}(y, x))] - \mathbb{E}_y[\min(0, -1 - D_{\text{SPD}}(y, G_{\text{SPD}}(y)))] \\
L_{G_{\text{SPD}}} &= -\mathbb{E}_y[D_{\text{SPD}}(y, G_{\text{SPD}}(y))] + \lambda_1 L_1^{\text{VGG}} + \lambda_2 L_1^{\text{Feat}},
\end{aligned}
\tag{2}
$$

where $G_{\text{SPD}}$, $D_{\text{SPD}}$ stand for the SPADE generator and discriminator, and $L_1^{\text{VGG}}$ and $L_1^{\text{Feat}}$ represent the VGG and discriminator feature matching $L_1$ loss functions, respectively (Park et al., 2019; Wang et al., 2018). We pre-train this network using pairs of real RGB images, $x$, and their corresponding real segmentation masks, $y$, from the target scene data set.

In the next section, we will describe how to employ the synthesized segmentation masks in an end-to-end manner to improve the performance of both the semantic bottleneck and the semantic image synthesis sub-networks.

## 3.3 END-TO-END FRAMEWORK

After training semantic bottleneck synthesis model to synthesize segmentation masks and the semantic image synthesis model to stochastically map segmentations to photo-realistic images, we adversarially fine-tune the parameters of both networks in an end-to-end approach by introducing an unconditional discriminator network on top of the SPADE generator (see Figure 2).

This second discriminator, $D_2$, has the same architecture as the SPADE discriminator, but is designed to distinguish between real RGB images and the fake ones generated from the *synthesized* semantic layouts. Unlike the SPADE conditional GAN loss, which examines pairs of input segmentations and output images, $(y, x)$ in equation 2, the GAN loss on $D_2$, $L_{D_2}$, is unconditional and only compares real images to synthesized ones, as shown in equation 3:

$$
\begin{aligned}
L_{D_2} &= -\mathbb{E}_x[\min(0, -1 + D_2(x))] - \mathbb{E}_z[\min(0, -1 - D_2(G(z)))] \\
L_G &= -\mathbb{E}_z[D_2(G(z))] + L_{G_{\text{SPD}}} + \lambda L_{G_{\text{SB}}}, \quad G(z) = G_{\text{SPD}}(G_{\text{SB}}(z))
\end{aligned}
\tag{3}
$$

where $G_{\text{SB}}$ represents the semantic bottleneck synthesis generator, and $L_{G_{\text{SB}}}$ is the improved WGAN loss to pretrain $G_{\text{SB}}$ described in Section 3.1. In contrast to the conditional discriminator in SPADE, which enforces consistency between the input semantic map and the output image, $D_2$ is primarily concerned with the overall quality of the final output. The hyper parameter $\lambda$ determines the ratio between the two generators during fine-tuning. The parameters of both generators, $G_{\text{SB}}$ and $G_{\text{SPD}}$, as well as the corresponding discriminators, $D_{\text{SB}}$ and $D_{\text{SPD}}$, are updated in this end-to-end fine-tuning.

We illustrate our final end-to-end network in Figure 2. Jointly fine-tuning the two networks in an end-to-end fashion allows the two networks to reinforce each other, leading to improved performance. The gradients with respect to RGB images synthesized by SPADE are back-propagated to the segmentation synthesis model, thereby encouraging it to synthesize segmentation layouts that lead to higher quality final images. Hence, SPADE plays the role of a loss function for synthesizing segmentations, but in the RGB space, hence providing a goal that was absent from the initial training. Similarly, fine-tuning SPADE with synthesized segmentations allows it to adapt to a more diverse set of scene layouts, which improves the quality of generated samples.

## 4 EXPERIMENTS AND RESULTS

We evaluate the performance of the proposed approach on two datasets containing images with complex scenes, where the ground truth segmentation masks are available during training (possibly only for a subset of the images). We also study the role of the two network components, semantic bottleneck and semantic image synthesis, on the final result. We compare the performance of SB-GAN against the state-of-the-art BigGAN model (Brock et al., 2019) as well as a ProGAN (Karras et al., 2017) baseline that has been trained on the RGB images directly. We evaluate our method using Fréchet Inception Distance (FID) as well as a perceptual evaluation.

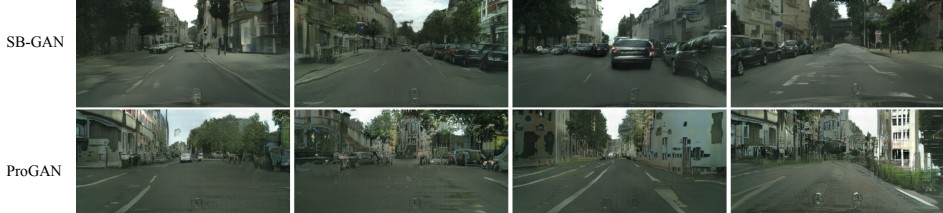

Figure 3: Images synthesized on Cityscapes-5K. Best viewed on screen; zoom in for more detail. Although both models capture the general scene layout, SB-GAN (1st row) generates more convincing objects, e.g. buildings and cars.

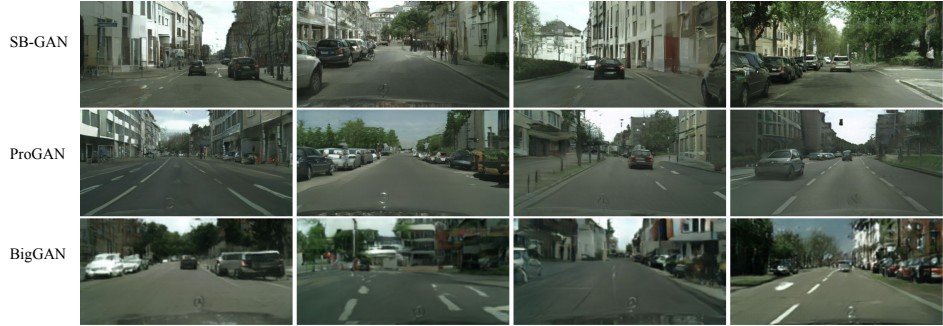

Figure 4: Images synthesized on Cityscapes-25K. Best viewed on screen; zoom in for more detail. Images synthesized by BigGAN (3rd row) are blurry and sometimes defective in local structures.

**Datasets** We study the performance of our model on the Cityscapes and ADE-indoor datasets as the two domains with complex scene images.

- Cityscapes-5K (Cordts et al., 2016) contains street scene images in German cities with training and validation set sizes of 3,000 and 500 images, respectively. Ground truth segmentation masks with 33 semantic classes are available for all images in this dataset.
- Cityscapes-25K (Cordts et al., 2016) contains street scene images in German cities with training and validation set sizes of 23,000 and 500 images, respectively with 19 semantic classes. Cityscapes-5K is a subset of this dataset, providing 3,000 images in the training set here as well as the entire validation set. Fine ground truth annotations are only provided for this subset, with the remaining 20,000 training images containing only coarse annotations. We extract the corresponding fine annotations for the rest of training images using the state-of-the-art segmentation model (Yu et al., 2017a) trained on the training annotated samples from Cityscapes-5K.
- ADE-Indoor is a subset of the ADE20K dataset (Zhou et al., 2017) containing 4,377 challenging training images from indoor scenes and 433 validation images with 95 semantic categories.

**Evaluation** We use the Fréchet Inception Distance (FID) (Heusel et al., 2017) as well as a perceptual evaluation of the quality of the generated samples. To compute FID, the real data and generated samples are embedded in a specific layer of a pre-trained Inception network. Then, a multivariate Gaussian is fit to the data, and the distance is computed as $\text{FID}(x, g) = ||\mu_x - \mu_g||_2^2 + \text{Tr}(\Sigma_x + \Sigma_g - 2(\Sigma_x \Sigma_g)^{\frac{1}{2}})$, where $\mu$ and $\Sigma$ denote the empirical mean and covariance, and subscripts $x$ and $g$ denote the real and generated data respectively. FID is sensitive to both the addition of spurious modes and to mode dropping (Sajjadi et al., 2018; Lucic et al., 2018). On the Cityscapes dataset, we ran five trials where we computed FID on 500 random synthetic images and 500 real validation images, and report the average score. On ADE-Indoor, this is repeated on batches of 433 images.

**Implementation details** In all our experiments, we set $\lambda_1 = \lambda_2 = 10$, and $\lambda = 10$. The initial generator and discriminator learning rates for training SPADE both in the pretraining and end-to-end steps are $10^{-4}$ and $4 \cdot 10^{-4}$, respectively. The learning rate for the semantic bottleneck synthesis sub-network is set to $10^{-3}$ in the pretraining step and to $10^{-5}$ in the end-to-end fine-tuning on

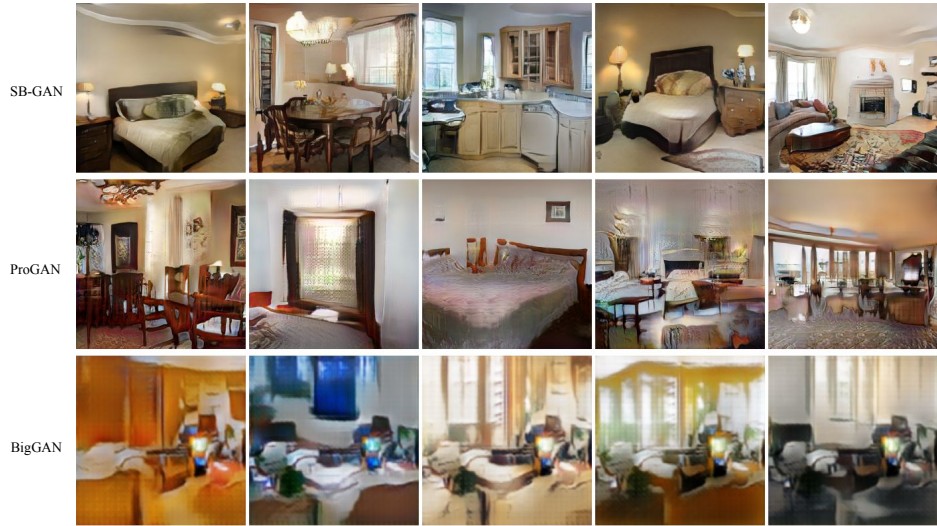

Figure 5: Images synthesized on ADE-Indoor. This dataset is very challenging, causing mode collapse for the BigGAN model (3rd row). In contrast, samples generated by SB-GAN (1st row) are generally of higher quality and much more structured than those of ProGAN (2nd row).

Table 1: FID of the synthesized samples (lower is better), averaged over 5 random sets of samples. Images were synthesized at resolution of $X \times 2X$ on Cityscapes and $X \times X$ on ADE-Indoor.

|  | (a) $X = 256$ | | | (b) $X = 128$ | | |
|---|---|---|---|---|---|---|
|  | ProGAN | SB-GAN W/O FT | SB-GAN | ProGAN | BigGAN | SB-GAN |
| CITYSCAPES-5K | 92.57 | 83.20 | **65.49** | 178.19 | - | **57.48** |
| CITYSCAPES-25K | 63.87 | 71.13 | **62.97** | 56.7 | 64.82 | **54.92** |
| ADE-INDOOR | 104.83 | 91.80 | **85.27** | 85.94 | 156.65 | **81.39** |

Cityscapes, and to $10^{-4}$ for ADE-Indoor. The temperature hyperparameter, $\tau$, is always set to 1. For BigGAN, we followed the setup by Lucic et al. (2019)[1], where we modified the code to allow for non-square images of Cityscapes. We used one class label for all images to have an unconditional BigGAN model. For both datasets, we varied the batch size (using values in $\{128, 256, 512, 2048\}$), learning rate, and location of the self-attention block. We trained the final model for 50K iterations.

## 4.1 QUALITATIVE RESULTS

In Figures 3, 4, and 5, we provide qualitative comparisons of the competing methods on the three aforementioned datasets. We observe that both Cityscapes-5K and ADE-Indoor are very challenging for the state-of-the-art ProGAN and BigGAN models, likely due to the complexity of the data and small number of training instances. Even at a resolution of $128 \times 128$ on the ADE-Indoor dataset, BigGAN suffers from mode collapse, as illustrated in Figure 5. In contrast, SB-GAN significantly improves the structure of the scene distribution and provides samples of higher quality. On Cityscapes-25K, the performance improvement of SB-GAN is more modest due to the large number of training images available. It is worth emphasizing that in this case only 3K ground truth segmentations are available to train SB-GAN. Compared to BigGAN, images synthesized by SB-GAN are sharper and contain more structural details (e.g., one can zoom-in on the synthesized cars). Additional synthesized semantic layouts and images are illustrated in Figures 7 to 10 in the Appendix.

---

[1]Configuration as in https://github.com/google/compare_gan/blob/master/example_configs/biggan_imagenet128.gin

Table 2: FID of the synthesized samples when conditioned on the ground truth labels. For SB-GAN, we train the entire model end-to-end and extract the trained SPADE.

|  | SPADE | SB-GAN |
|---|---|---|
| CITYSCAPES-5K | 72.12 | **60.39** |
| CITYSCAPES-25K | 60.83 | **54.13** |
| ADE-INDOOR | 50.30 | **48.15** |

Table 3: Average perceptual evaluation scores when each evaluators has selected a quality score in the range of 1 (terrible quality) to 4 (high quality) for each image.

|  | ProGAN | BigGAN | SB-GAN |
|---|---|---|---|
|  | 2.08 | - | **2.48** |
|  | 2.53 | 2.27 | **2.61** |
|  | 2.35 | 1.96 | **2.49** |

## 4.2 QUANTITATIVE EVALUATION

To provide a thorough empirical evaluation of the proposed approach, we generate samples for each dataset and report the FID scores of the resulting images (averaged across 5 sets of generated samples). We evaluate SB-GAN both before and after end-to-end fine-tuning, and compare our method to two strong baselines, ProGAN (Karras et al., 2017) and BigGAN (Brock et al., 2019).

The results are detailed in Tables 1a and 1b. First, in the low-data regime, even without fine-tuning, our Semantic Bottleneck GAN produces higher quality samples and significantly outperforms the baselines on Cityscapes-5K and ADE-Indoor. The advantage of our proposed method is even more striking on smaller datasets. While competing methods are unable to learn a high-quality model of the underlying distribution without having access to a large number of samples, SB-GAN is less sensitive to the number of training data points. Secondly, we observe that by jointly training the two components, SB-GAN produces state-of-the-art results across all three datasets.

We were not able to successfully train BigGAN at a resolution of $256 \times 512$ due to instability observed during training and mode collapse. Table 1b shows the results for a lower-resolution setting, for which we were able to successfully train BigGAN. We report the results before the training collapses. BigGAN is, to a certain extent, able to capture the distribution of Cityscapes-25K, but fails completely on ADE-Indoor. Interestingly, BigGAN fails to capture the distribution of Cityscapes-5K even at $128 \times 128$ resolution.

**Generating by conditioning on real segmentations**   To independently assess the impact of end-to-end training on the conditional image synthesis sub-network, we evaluate the quality of generated samples when conditioning on ground truth validation segmentations from each dataset. Comparisons to the baseline network SPADE (Park et al., 2019) are provided in Table 2 and Figures 13 and 14 in the Appendix. We observe that the image synthesis component of SB-GAN consistently outperforms SPADE across all three datasets, indicating that fine-tuning on data sampled from the segmentation generator improves the conditional image generator.

**Fine-tuning ablation study**   To dissect the effect of end-to-end training, we perform a study on different components of SB-GAN in the Appendix. In particular, we consider three settings: (1) SB-GAN before end-to-end fine-tuning, (2) fine-tuning only the semantic bottleneck synthesis component, (3) fine-tuning only the conditional image synthesis component, and (4) fine-tuning all jointly.

## 4.3 PERCEPTUAL EVALUATION

We used Amazon Mechanical Turk (AMT) to assess the performance of each method on each dataset using ~600 pairs of (synthesized images, human evaluators) with a total of 200 unique synthesized images. For each image, evaluators were asked to assign a score between 1 to 4 to each image, indicating low-to-high quality images, respectively. The results are summarized in Table 3 and are consistent with our FID-based evaluations.

## 5 CONCLUSION

We proposed an end-to-end Semantic Bottleneck GAN model that synthesizes semantic layouts from scratch, and then generates photo-realistic scenes conditioned on the synthesized layouts. Through extensive quantitative and qualitative evaluations, we showed that this novel end-to-end training pipeline significantly outperforms the state-of-the-art models in unconditional synthesis of complex

scenes. In addition, Semantic Bottleneck GAN strongly improves the performance of the state-of-the-art semantic image synthesis model in synthesizing photo-realistic images from ground truth segmentations. As a future work, one could explore novel ways to train GANs with discrete outputs, especially to deal with the non-differentiable nature of the generated outputs.

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

## A  APPENDIX

### A.1  DESIGN DIFFERENCES OF SB-GAN AND TEXT-TO-IMAGE MODELS INFERRING SEMANTIC LAYOUT

Here, we clarify differences between our proposed model and the conditional text-to-image synthesis models inferring a semantic layout. As illustrated in Figure 6-(b), the text-to-image synthesis models proposed by Hong et al. (2018) and Li et al. (2019) decompose the synthesis task into multiple steps. Given the text description, a semantic layout is constructed by generating object bounding boxes and refining each box by estimating object shapes. Then, an image is synthesized conditionally on the generated semantic layout from the first step. We note that the above approaches, as opposed to ours, are conditional and require supervision in the form of textual descriptions. Secondly, they are sequential in nature and synthesize masks of a few objects (e.g. person, elephant), but not a fully fine-grained semantic map (e.g. missing sky, grass, etc.). In stark contrast, our approach unconditionally and progressively synthesizes the *full semantic layout* of the entire scene from a noise input in an end-to-end network design as shown in Figure 6-(a).

### A.2  ADDITIONAL RESULTS

In Figures 7, 8, 9, and 10, we show additional synthetic results from our proposed SB-GAN model including both the synthesized segmentations and their corresponding synthesized images from the Cityscapes-25K and ADE-Indoor datasets. As mentioned in the paper, on the Cityscapes-25K dataset, fine ground truth annotations are only provided for the Cityscapes-5k subset. We extract

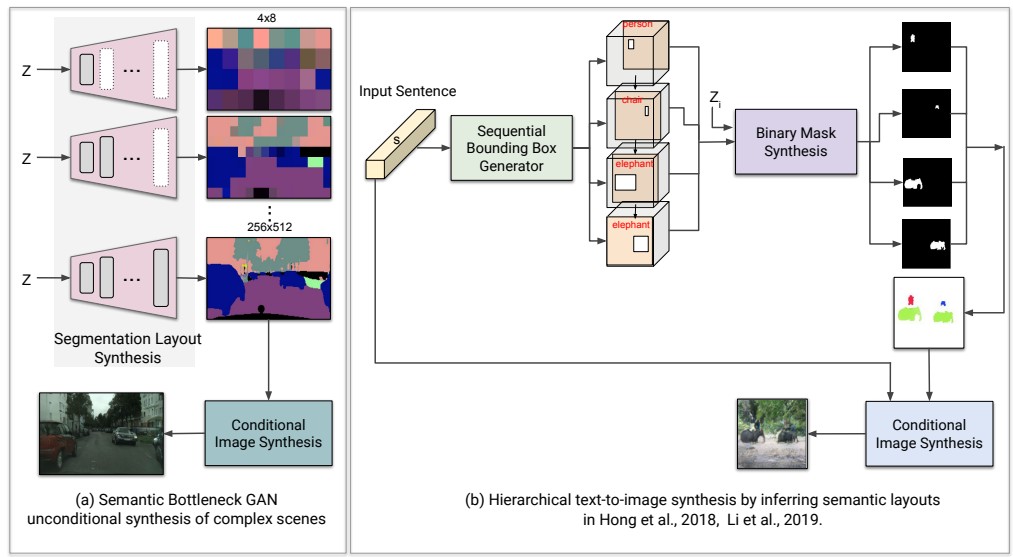

Figure 6: Architectural differences between our unconditional semantic bottleneck synthesis network and the conditional semantic layout synthesis network in Hong et al. (2018) and Li et al. (2019). (a) Schematic of our unconditional Semantic Bottleneck GAN. We progressively train an adversarial segmentation synthesis network to generate realistic segmentation maps from scratch, then synthesize a photo-realistic image using a conditional image synthesis network. End-to-end coupling of these two components results in state-of-the-art unconditional synthesis of complex scenes. For more detail about our conditional image synthesis network, one can refer to Section 3.2. (b) Schematic of the hierarchical text-to-image synthesis models inferring a semantic layout (Hong et al., 2018; Li et al., 2019). From an encoding of the input sentence, object bounding boxes are generated sequentially using an auto-regressive decoder, and are refined by a synthesized binary shape mask in the next step. The final image is synthesized given the constructed semantic layout and the text description. Note that whereas (b) conditionally generates masks only for objects, our model (a) unconditionally generates segmentation maps for the entire scene.

the corresponding fine annotations for the rest of training images using the state-of-the-art segmentation model Yu et al. (2017a); Yu & Koltun (2016) trained on the training annotated samples from Cityscapes-5K.

**Fine-tuning ablation study**   To further dissect the effect of end-to-end training, we perform a study on different components of SB-GAN. In particular, we consider three settings: (1) SB-GAN before end-to-end fine-tuning, (2) fine-tuning only the semantic bottleneck synthesis component, (3) fine-tuning only the conditional image synthesis component, and (4) fine-tuning all components jointly. The results on the Cityscapes-5K dataset (resolution $128 \times 256$) are reported in Table 4. Finally, the impact of fine-tuning on the quality of samples can be observed in Figures 11 and 12.

Table 4: Ablation study of various components of SB-GAN. We report FID scores of SB-GAN before fine-tuning, fine-tuning only the semantic bottleneck synthesis component, fine-tuning only the image synthesis component, and full end-to-end fine-tuning. Experiments are performed on the Cityscapes-5K dataset at a resolution of $128 \times 256$.

| No FT | FT SB | FT SPADE | FT Both |
|-------|-------|----------|---------|
| 70.15 | 66.22 | 63.04 | **58.67** |

**Generating by conditioning on real segmentations**   Figures 13 and 14 present additional examples illustrating the impact of SB-GAN on improving the performance of SPADE (Park et al., 2019), the state-of-the-art semantic image synthesis model on ground truth segmentations. The third row

Synthesized Segmentations          Synthesized Images

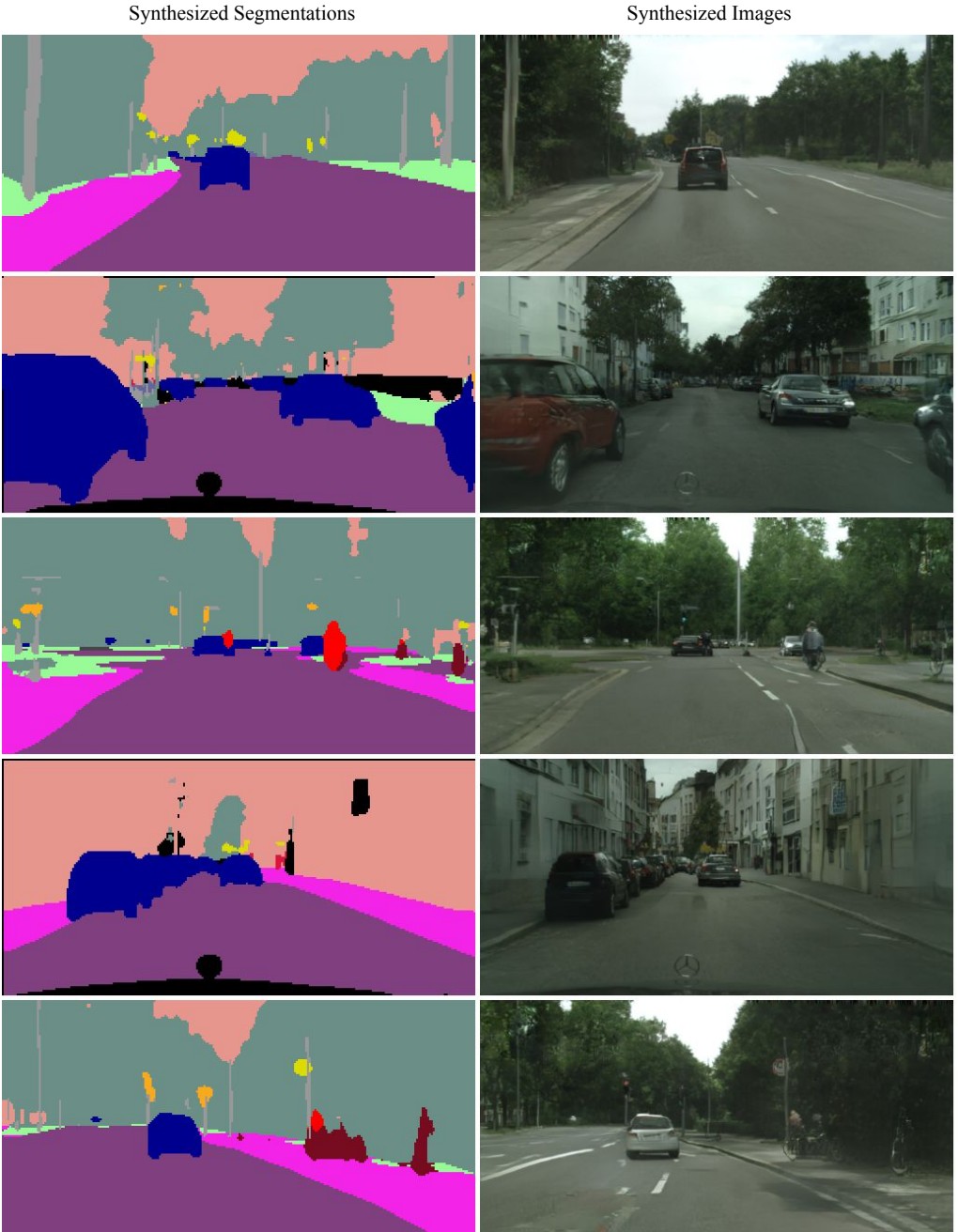

Figure 7: Segmentations and their corresponding images synthesized by SB-GAN trained on the Cityscapes-25K dataset.

in these two figures show examples of the synthesized images conditioned on ground truth labels when the SPADE sub-network is extracted from a trained SB-GAN model.

Synthesized Segmentations            Synthesized Images

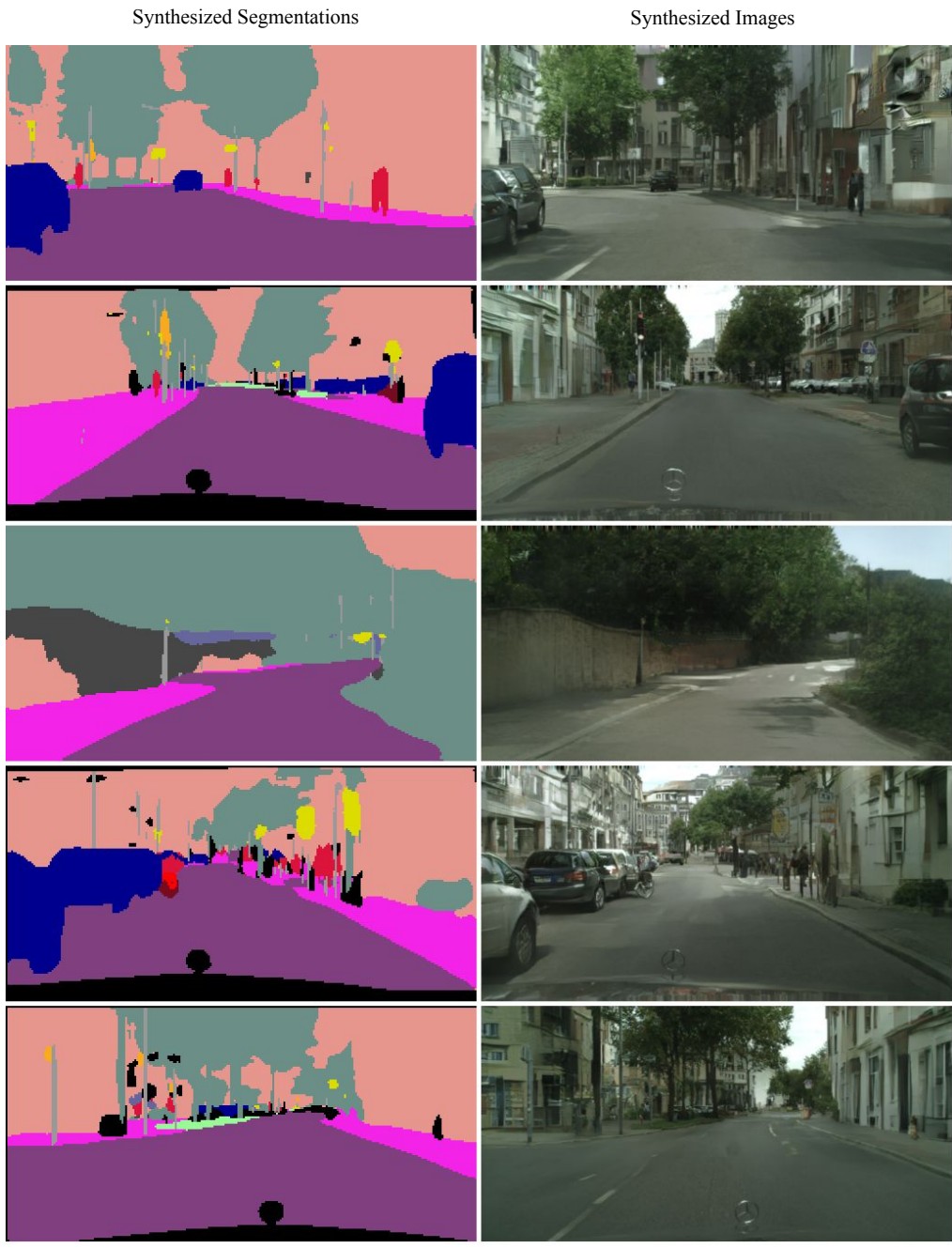

Figure 8: Segmentations and their corresponding images synthesized by SB-GAN trained on the Cityscapes-25K dataset.

Synthesized Segmentations   Synthesized Images   Synthesized Segmentations   Synthesized Images

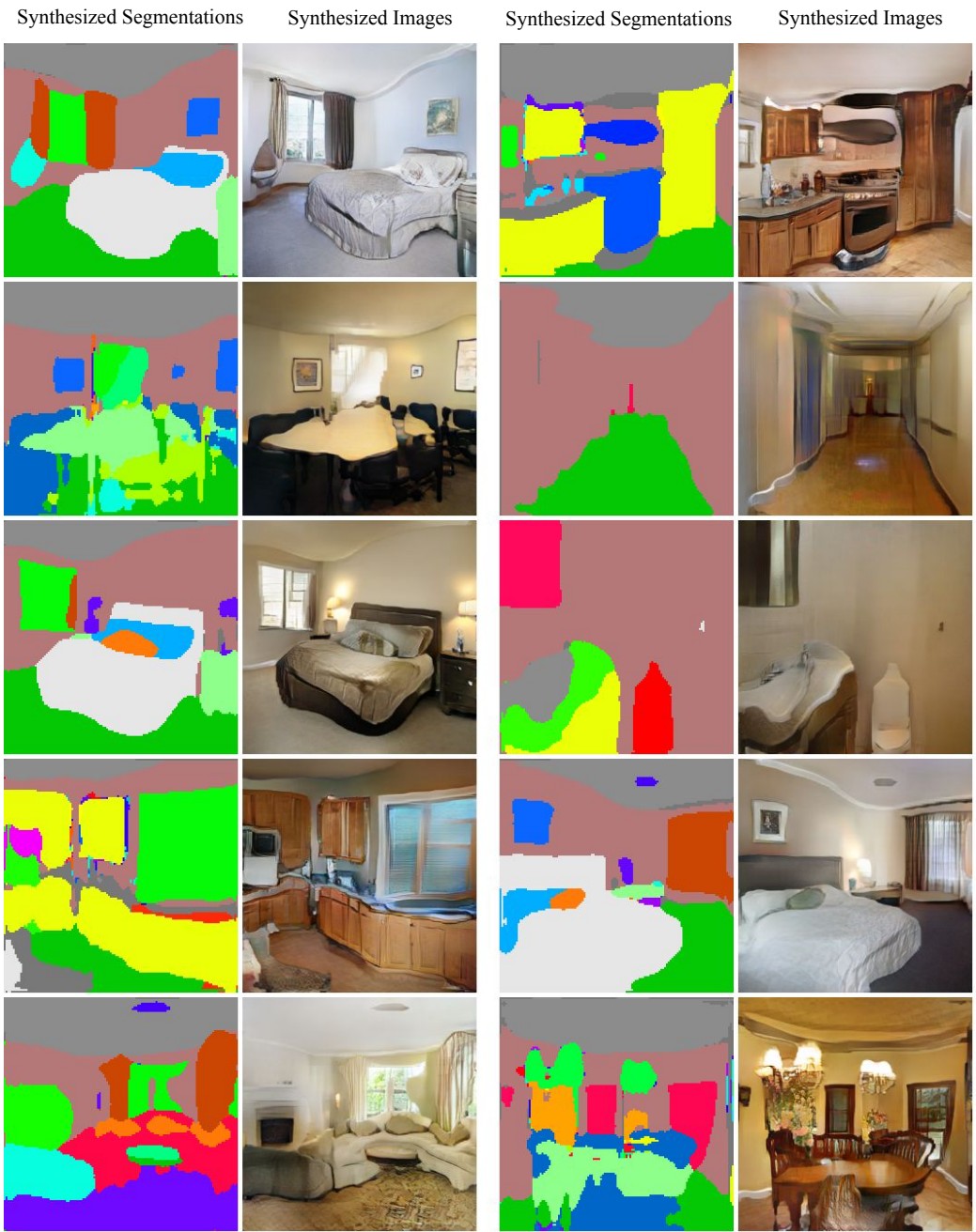

Figure 9: Segmentations and their corresponding images synthesized by SB-GAN trained on the ADE-Indoor dataset.

Synthesized Segmentations    Synthesized Images    Synthesized Segmentations    Synthesized Images

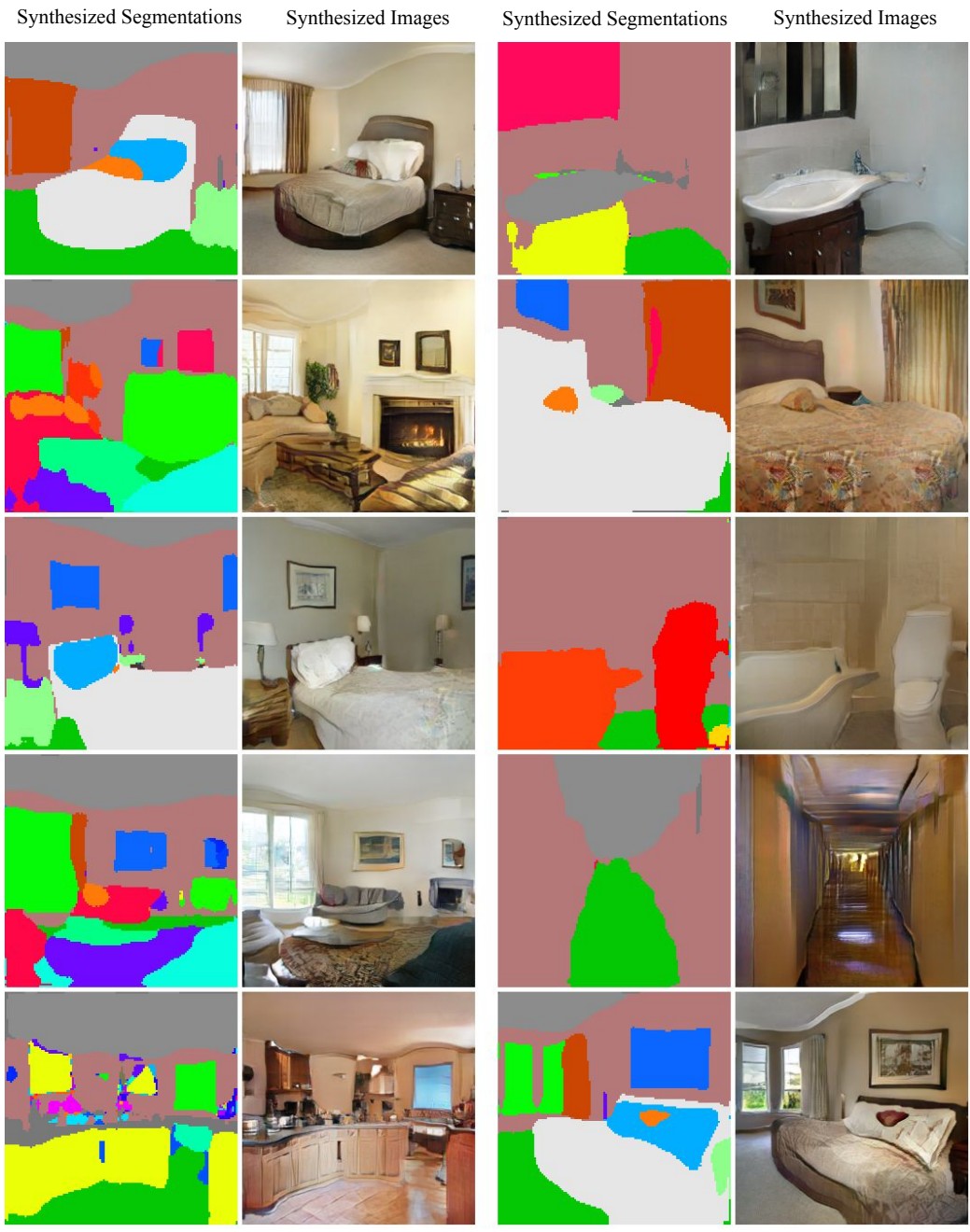

Figure 10: Segmentations and their corresponding images synthesized by SB-GAN trained on the ADE-Indoor dataset.

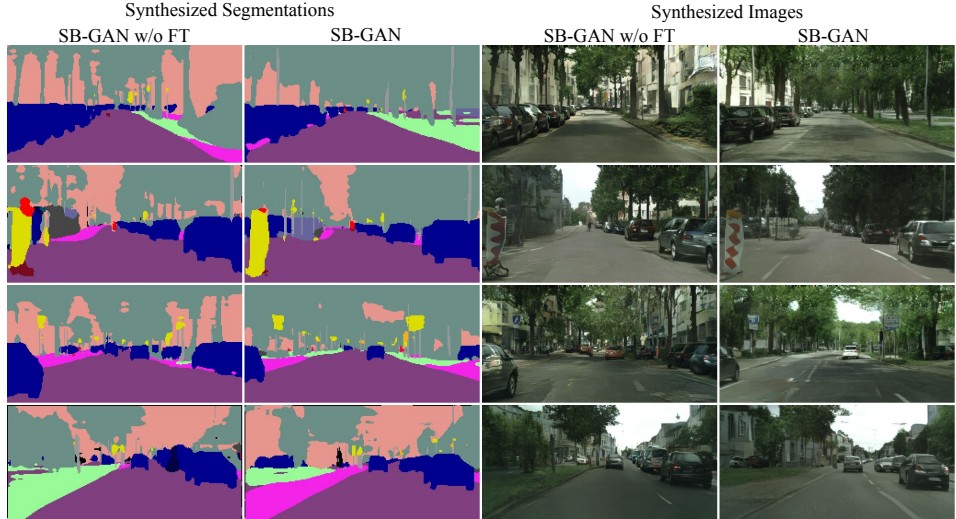

Figure 11: The effect of fine-tuning on the baseline setup for the Cityscapes-25K dataset. We observe improvements in both the global structure of the segmentations and the performance of semantic image synthesis, resulting in images of higher quality.

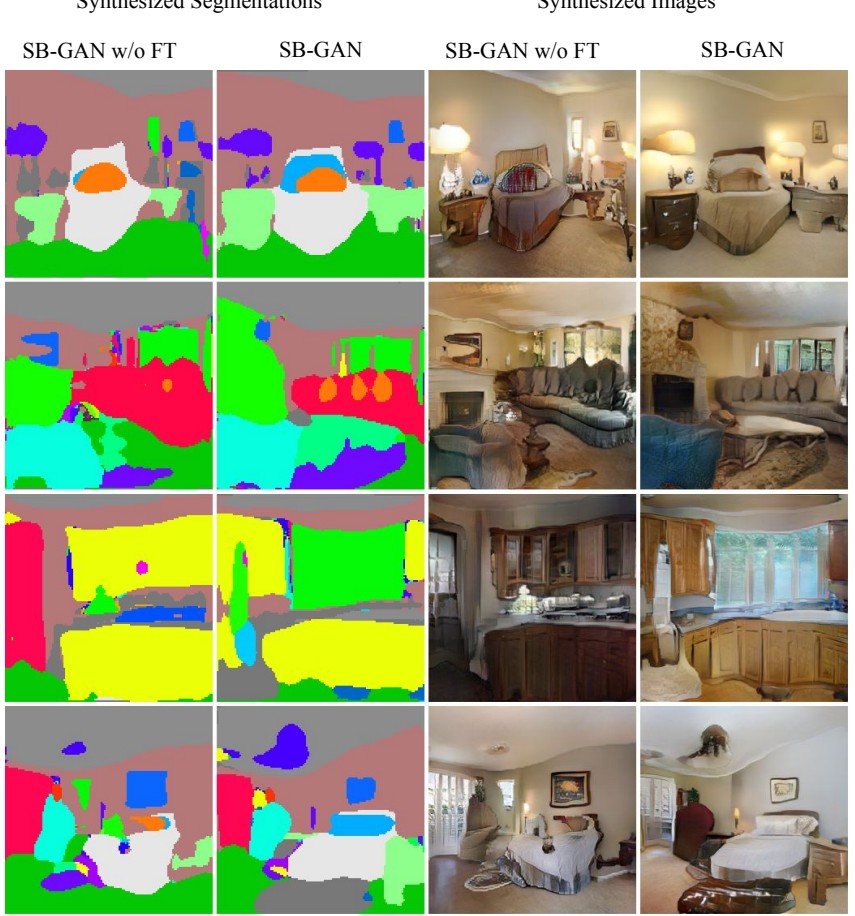

Figure 12: The effect of fine-tuning (FT) on the baseline setup for ADE-Indoor dataset. Analogously to the results on Cityscapes-25K, we observe improvements in both the global structure of the segmentations and the performance of semantic image synthesis.

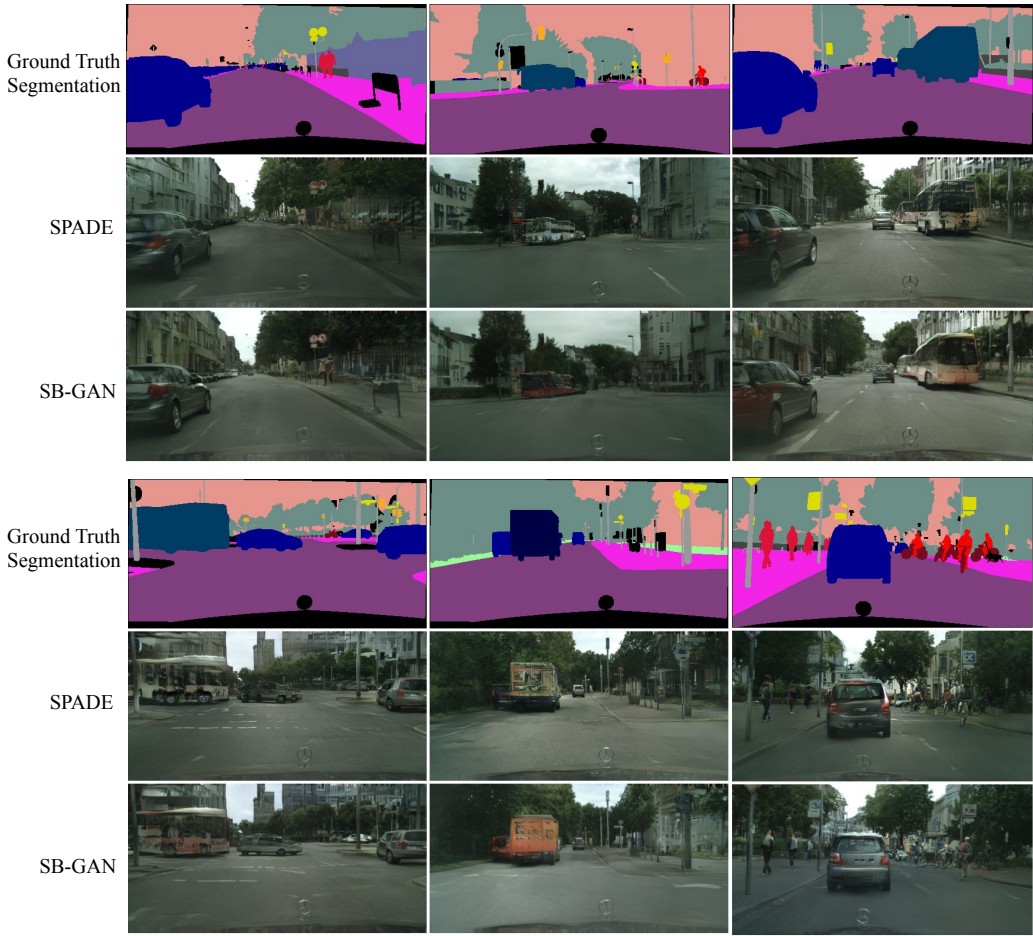

Figure 13: The effect of SB-GAN on improving the performance of the state-of-the-art semantic image synthesis model (SPADE) on ground truth segmentations of Cityscapes-25K validation set. For SB-GAN, we train the entire model end-to-end, extract the trained SPADE sub-network, and synthesize samples conditioned on the ground truth labels.

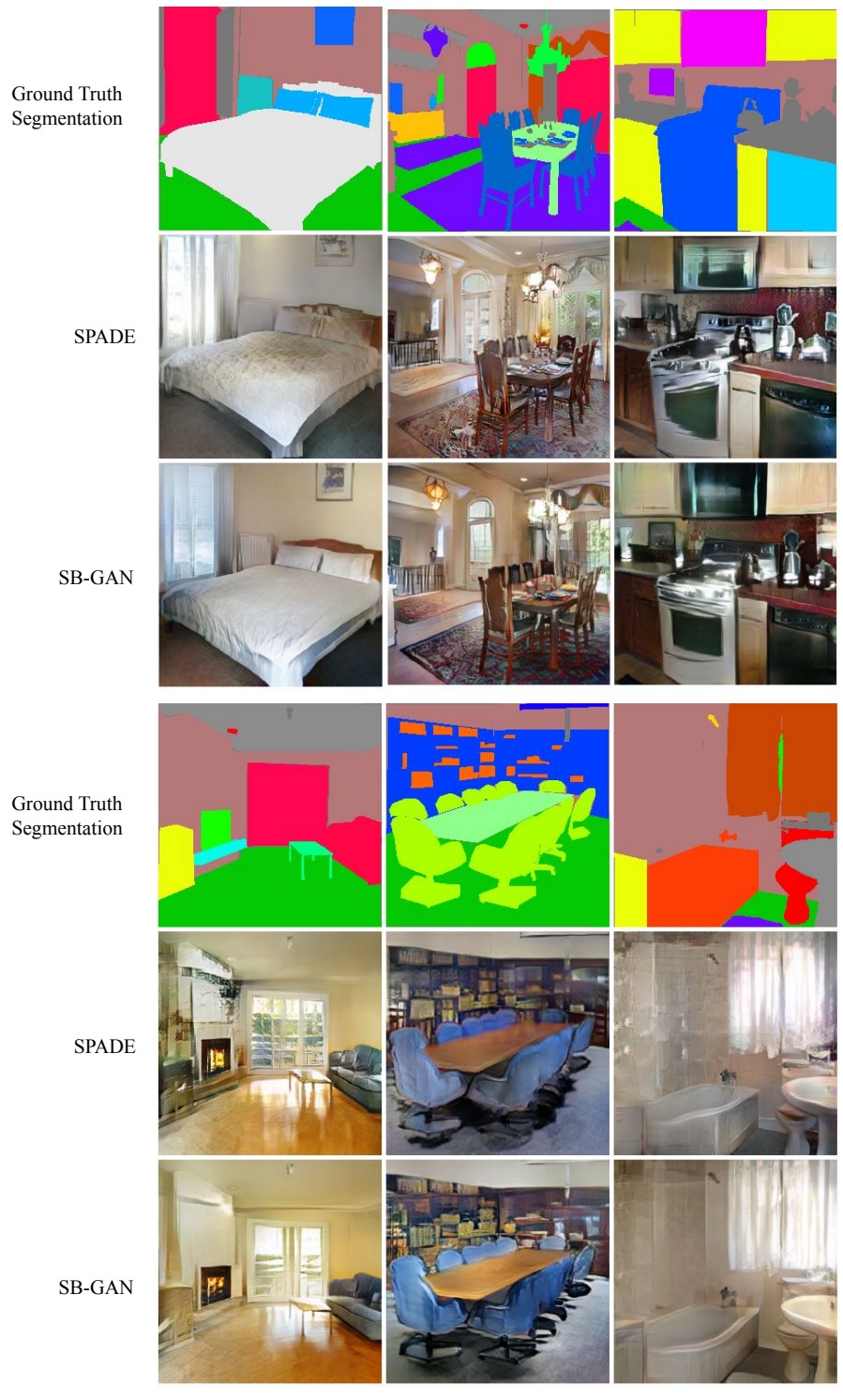

Figure 14: The effect of SB-GAN on improving the performance of the state-of-the-art semantic image synthesis model (SPADE) on ground truth segmentations of ADE-Indoor validation set. For SB-GAN, we train the entire model end-to-end, extract the trained SPADE sub-network, and synthesize samples conditioned on the ground truth labels.

