# OpenReview forum: "Unconditional Synthesis of Complex Scenes Using a Semantic Bottleneck"
_ICLR.cc/2021/Conference — Reject_

### Official Review · AnonReviewer2 · 2020-10-21
**Need more evidence the semantic bottleneck is not a blocker for practical usage**

**Rating:** 6
**Confidence:** 4

**Review:**

1. Summary. This paper considers a two-stage procedure of generation complex scenes (cityscapes or living rooms, i.e. without a central object): Firstly, a noise vector is mapped to a discrete semantic map (Gumbel softmax and straight-through estimator are applied). Secondly, the obtained segmentation map is translated to an RGB image using the SPADE architecture. Both networks are pretrained separately and then finetuned in an end-to-end manner.

1. Decision. What I really like about this work is the attempt to dissect the scene generation into several steps. This may be of great interest in the case of datasets of limited size, as the authors showcase. What is more, this work tackles the problem of the generation of discrete semantic labels and has success.  However, the chosen intermediate representation is, to my mind, the main concern of the presented approach as ground-true semantic labels are not so easy to crowdsource. This makes the considered pipeline a bit limited in practical usage. A possible solution is to employ the off-the-shelf pretrained segmentation net (as in your Cityscapes-25k experiment): I believe the paper would be stronger with an additional experiment on more common datasets for scene generation (e.g., LSUN bedrooms), labeled with such a network. One more thing, I would recommend choosing a more recent and appealing baseline like StyleGAN instead of ProGAN. Summarizing, I tend to vote for accepting, though believe that the experiments could be more solid.

1. Questions
    1. Is the sampling step with the Gumbel-softmax trick crucial for the generation of semantic maps? Why the combination of simple softmax scores and straight-through estimator does not suffice?
    1. How exactly do you downscale the ground-true segmentation maps to feed the discriminator at coarse scales of progressive learning?

---

> ### Author Response · Authors · 2020-11-24
> **Thanks for your review!**
>
> Thanks very much for your time and feedback.
>
> **Run on more common datasets:**
> Thanks for the suggestion. As you mentioned, the focus of this work is more on complex datasets with limited size but this is an interesting experiment to be explored.
>
> **StyleGAN as a baseline:**
> As you suggested, we have trained StyleGAN on the ADE_indoor dataset with 4377 images, and after several hyperparameter tuning runs, the best FID score that we get at the resolution of 256x256 is 230 and that at the resolution of 128x128 is 179 (which is significantly worse to the ones reported in Tables 1 and 2, respectively.) Since StyleGAN is designed for controllable disentanglement setups, its poor performance on ADE_indoor could be due to the complexity of this small dataset for such a disentanglement (note that StyleGAN was trained on scenes with a single central salient object such as a face, a car, or a bed). Moreover, the StyleGAN repository does not support rectangular images and our attempts to modify it within this limited timeline were not fruitful to evaluate it on Cityscapes-5K and Cityscapes-25K. We will add the results once we can successfully train the model on these two datasets as well. Overall, as discussed in the paper, the existing SOTA models in unconditional image generation fail when the scene is complex and the dataset is small.
>
> **Why Gumble-softmax?**
> We had tried training the model with Softmax combined with a  straight-through estimator. However, we observed that using a Gumbel softmax instead improves the stability of the training model. Intuitively, Gumbel softmax provides a more accurate estimation of the categorical distribution of per-pixel segmentation labels while softmax predictions are less accurate during training which can destabilize the training, especially in the initial iterations.
>
> **Downscaling ground-truth Segmentations:**
> We use the nearest neighbor-based resize transformation to rescale the semantic layouts at each resolution.

---

### Official Review · AnonReviewer3 · 2020-10-28
**This work introduces an interesting approach that intelligently weaves two networks for unconditional label generation and label-conditioned image synthesis. Experimental results support its superior performance.**

**Rating:** 8
**Confidence:** 4

**Review:**

This work proposes a novel approach for unconditioned image synthesis of complex scenes by intelligently coupling two major tasks; unconditional label generation and label-conditioned image synthesis. To overcome the limitation of failing to generate high-fidelity complex scenes using current GAN-based approaches, this method proposes to divide this into two parts: unconditional segmentation map synthesis network and conditional segmentation-to-image synthesis network. The former is based on the ProGAN with some modifications in losses to deal with discrete semantic labels, and the latter leverages the existing method (Park et al. 2019) based on SPADE residual blocks. Experimental results demonstrate superior performance on the complex scene synthesis. Additionally, the latter part for segmentation-to-image synthesis task also outperforms the existing method (Park et al. 2019) thanks to joint end-to-end training with the former using ProGAN.

* Pros
1) Decomposing the complex scene synthesis into two sub-tasks (segmentation map generation and segmentation-to-image synthesis) looks novel, also validating outstanding performance over SOTA.
2) Segmentation-to-image synthesis is also boosted, thanks to joint end-to-end training with the unconditional segmentation map synthesis network.

* Cons
1) Though this paper is well-written, some parts need more details.
- In Section 3.1, it would be nice to explain how to generate semantic segmentation maps progressively by referring to Figure 1.
- Eq (3) uses loss functions from two sub-networks (semantic bottleneck synthesis network and semantic image synthesis network).
Why did you not use L_D_SPD and L_G_SPD in (2) for training the whole network with a pair of real RGB images and real segmentation maps.
It seems that L_1^VGG and L_1^Feat can also be used in (3).

- Semantic bottleneck synthesis in Section 3.1 needs more explanations, e.g., how to convert real segmentation maps into probability maps, using argmax and soft argmax in forward and backward passes.

---

> ### Author Response · Authors · 2020-11-24
> **Thanks for your review!**
>
> Thanks very much for your time and feedback.
>
> **Progressive generation of segmentation maps:**
> Thanks for your comment, we have clarified this in Sec 3.1.
>
> **Eq3:**
> $L_{G_{SPD}}$ defined in Eq. 2 is included as a loss term in $L_G$ in Eq.3 while training the model end-to-end. This loss term contains the conditional SPADE GAN loss in addition to the VGG and feature matching loss functions.
>
> **Convert real segmentation maps to probability maps:**
> We do not change the format of real segmentation maps during the GAN training. We synthesize segmentation layouts by first generating per-pixel probability scores of belonging to each of the K semantic classes and then sampling a semantic class per pixel. Similar to the real segmentation layout samples, the synthesized maps fed to the discriminator should contain discrete per-pixel semantic categories. They should be also differentiable to enable back-propagating to the generator in the GAN training. We use the Gumbel Softmax + straight-through estimation to enable differentiability in the backward pass while containing discrete labels in the forward pass. We have clarified this in Sec 3.1.

---

### Official Review · AnonReviewer4 · 2020-10-28
**Reasonable approach but validation is unconvincing**

**Rating:** 4
**Confidence:** 4

**Review:**

The paper suggests a new approach for unconditional generation of complex scenes. The approach performs generation in two steps: first a semantic map is generated from noise using a conventional generator architecture, then the semantic image is turner into an RGB image by SPADE translator.

The paper has several strengths. First, the idea is clear and may make sense (though this has not been shown convincingly). Furthermore, the paper is well written and has detailed related work review (though some important papers are missed). The results are also interesting.

Despite the strenghts, I think that the paper may not be suitable for ICLR in the current form for the following reasons:

1) Novelty. The idea of two stage generation of complex images with GANs has been proposed before in a well-known paper [Wang & Gupta, Generative Image Modeling Using Style and Structure Adversarial Networks, ECCV16] . There, the image of normals rather than semantic segmentation served as an intermediate ("bottleneck") representation, otherwise the idea is very similar. It is likely that the normal map may be a better intermediate representation since it is continuous-valued and does not need to deal with discretization issues. Overall, a comparison and a proper positioning w.r.t. [Wang&Gupta] is needed.

2) Deficient comparisons with 1-step GANs. The authors for some reason chose Progressive GANs and BigGAN as the reference 1-stage GANs. This choice is totally unclear to me. StyleGAN v1 and v2 are improved versions of ProGAN and should have been tried instead. Given the use of SPADE (i.e. style based generator) in the authors' architecture, the comparison to StyleGAN would be all the more natural. I find it very likely that the result of StyleGAN can be very similar or better than the authors after similar amount of tuning, especially on Cityscapes 25K. I am therefore not convinced that the proposed idea is actually working.

3) [Minor] There is a published CVPR workshop paper with a very similar idea https://openaccess.thecvf.com/content_CVPRW_2020/html/w23/Volokitin_Decomposing_Image_Generation_Into_Layout_Prediction_and_Conditional_Synthesis_CVPRW_2020_paper.html . The results are worse and some important differences exist. However, it does undermine the novelty. Still a CVPR-workshop paper may be missed by the community, so I weigh this issue as minor.

4) The results are interesting, but they are not terrific. I wonder if the authors can scale their method to higher resolutions (which would be very useful for complex scenes), or if it would break down in some way.

5) [Minor; suggestion not a criticism] There is an obvious use for what the authors are doing. The data they generate can be used to train semantic segmentation networks (essentially serving as dataset augmentation). I think having the evaluation of this aspect would be useful and would make the paper stronger.

6) [Minor] The phrase "In fact, due to the missing ordering relationships, generating smooth
segmentation maps cannot be enforced by smoothness among values of neighboring pixels." should be reformulated. A large body of work exist (generally associated with MRF/CRFs in computer vision; and also in statistical physics) on enforcing smoothness (in different senses) in discrete-valued maps, e.g. via Potts prior.

To sum up, the idea is clear and well described, but the paper does not convince me that the idea is working (improves over StyleGAN; can synthesize high-res images) and in particular that semantic bottleneck is working better than other bottlenecks (e.g. proposed by [Wang&Gupta]).

---

> ### Author Response · Authors · 2020-11-24
> **Thanks for your review!**
>
> Thanks very much for your time and feedback.
>
> **Novelty:**
> Thanks for bringing Wang et al [2016] to our attention. Both segmentation layouts and surface normals capture the structure and provide a coarse estimation of a scene. However, a semantic bottleneck has several advantages in an unconditional image synthesis problem as discussed below:
>
> (i) Ground Truth Data: Providing ground truth depth or surface normal is arguably more problematic for an arbitrary dataset than segmentation labels. While it is feasible to acquire ground truth segmentation labels via labeling jobs on the cloud, providing depth images for an arbitrary dataset is infeasible unless already computed by a depth camera. Ground truth depth images can be also “estimated” through a depth estimation network that has been trained on a different dataset, but this could lead to inaccurate images due to the distribution shift between different domains. This problem is however less severe for segmentations since a segmentation model can be either trained or fine-tuned on a small set of segmentation labels obtained from the labeling job on the cloud, as we demonstrate in our experiments on Cityscapes-25K.
>
> (ii) Structure Synthesis: Although normal maps are continuous-valued and pass the gradients more easily through the end-to-end image synthesis network, their synthesis could be more complex since object-level information is missing. Every single object is decomposed as smaller segments with different surface normals while the object’s surface normal in each dimension is equal to the surface normal of a few other objects in the scene.
>
> (iii) Transforming intermediate representation to an image: The rapid progress in the label conditional image synthesis models (compared with the surface normal conditional models) proves the important role and simplicity of using segmentation layouts for high-fidelity natural image synthesis. Coupled with an unconditional segmentation synthesis model, it makes our full unconditional model flexible to benefit from the future improvements in segmentation-to-image synthesis approaches. For instance, the SPADE technique we used in our segmentation-to-image synthesis model uses semantic labels to modulate the batch normalized activations spatially while it is unclear how suitable SPADE is for surface normals (this could be a future research direction to adapt SPADE to surface normals).
>
> Given the pros and cons of these two intermediate representations, it would be an interesting future research direction to use both segmentation and surface normal maps simultaneously as the bottleneck in an unconditional image synthesis pipeline.
>
> We also thank the reviewer for reminding us about Volokitin et al [CVPR/W 2020]. This paper was developed as a concurrent work. As you noted, there are substantial differences in these works, and our results are much more compelling both quantitatively and qualitatively.
> Thank you for your suggestion, we have added the above papers to our related works section.
>
> **StyleGAN as a baseline:**
> As you suggested, we have trained StyleGAN on the ADE_indoor dataset with 4377 images, and after several hyperparameter tuning runs, the best FID score that we get at the resolution of 256x256 is 230 and that at the resolution of 128x128 is 179 (which is significantly worse to the ones reported in Tables 1 and 2, respectively.)  Since StyleGAN is designed for controllable disentanglement setups, its poor performance on ADE_indoor could be due to the complexity of this small dataset for such a disentanglement (note that StyleGAN was trained on scenes with a single central salient object such as a face, a car, or a bed).
> Moreover, the StyleGAN repository does not support rectangular images and our attempts to modify it within this limited timeline were not fruitful to evaluate it on Cityscapes-5K and Cityscapes-25K.  We will add the results once we can successfully train the model on these two datasets as well. Overall, as discussed in the paper, the existing SOTA models in unconditional image generation fail when the scene is complex and the dataset is small.
>
> **Higher resolution:**
> The cityscapes images synthesized by our model are at the resolution of 256x512 which is already a high resolution. There is no limitation for our model to be trained at a higher resolution but the available compute resources limited us to this resolution. Moreover, it is worth noting that our baselines, especially BigGAN, break down even at the resolution of 128x256 on these complex scenes revealing the out-performance and stability of our proposed model.
>
> **Use for data augmentation:**
> Thank you for the suggestion. We agree that our method can be very useful in adding extra training data points for segmentation models, and it would be an interesting research direction to be explored in the future.
>
> **Rephrase segmentation smoothness:**
> Thank you for your suggestion, we have fixed this in the paper.

---

### Official Review · AnonReviewer1 · 2020-10-29
**Review for SB-GAN**

**Rating:** 6
**Confidence:** 5

**Review:**

In this paper, the authors propose a new paradigm for unconditional image synthesis with semantic layouts as the bottleneck. The presented approach is straightforward: we can first sample a semantic layout from a latent variable, and then perform image synthesis from this semantic layout. The proposed method is able to synthesize images that look more realistic than unsupervised image synthesis methods such as BigGAN.

Strengths: - The idea of having some predefined intermediate representation (such as semantic layout) can help improve unconditional image synthesis.
- The approach works well on the datasets with semantic layouts, including Cityscapes and ADE20K datasets.
- The experiments are comprehensive and convincing.
- The proposed end-to-end fine-tuning can improve the visual results.

Weaknesses:
- Apparently, there is a limitation of the proposed approach: it can only work on datasets with semantic layouts.
- This overall idea of having semantic layouts as a bottleneck is simple. Technically, semantic bottle synthesis and semantic image synthesis are based on well-known models (ProGAN+WGAN, SPADE). The technical contributions may appear limited, but I appreciate the overall idea for unconditional image synthesis.
- Why the perceptual evaluation of BigGAN in Table 3 for Cityscapes-5K is not available?

---

> ### Author Response · Authors · 2020-11-24
> **Thanks for your review!**
>
> Thanks very much for your time and feedback.
>
> **Requiring semantic layout:**
> As discussed in Sec. 4, we have studied the role of imperfect segmentations in the Cityscapes-25K dataset where fine pixel annotations (segmentation labels) of only 3K images, i.e., the training set of Cityscapes-5K, are provided.  We extracted the corresponding segmentation maps for the rest of the training images using the DRN segmentation model [Yu et al, 2017a] trained on the 3K training annotated samples from Cityscapes-5K. The results are analyzed in Fig. 4 as well as Tables 1 and 2, revealing that SB-GAN outperforms the baseline models even when a small fraction of the training images are paired with their ground-truth segmentations. We also note that semantic labels for a small set of images in an arbitrary dataset can be easily acquired on the cloud via labeling jobs. This answers the concern about the capability of SB-GAN in the absence of high-quality segmentations.
>
> **Novelty:**
> Dividing the unconditional synthesis of complex scenes into two steps to benefit from (i) high-fidelity generation of label conditional models and (ii) the flexibility of unconditional image generation is by itself a novel approach. While both unconditional generation and image-to-image translation are well-explored learning problems, fully unconditional generation of the segmentation maps is a hard task: (i) Semantic categories do not respect any ordering relationships and the network is therefore required to capture the intricate relationship between segmentation classes, their shapes, and their spatial dependencies. (ii) As opposed to RGB values, semantic categories are discrete, hence non-differentiable which poses a challenge for end-to-end training (Sec. 3.2). (iii) Naively combining state-of-the-art unconditional generation and image-to-image translation models leads to poor performance. However, by carefully designing an additional discriminator component and a corresponding training protocol, we not only manage to improve the performance of the end-to-end model, but also the performance of each component separately (sec 3.3).
>
> **BigGAN for Cityscapes5k in Table 3:**
> BigGAN is, to a certain extent, able to capture the distribution of Cityscapes-25K, but fails completely on Cityscapes-5K even at the resolution of 128x128. Our best run reached FID close to 200 at around 50k steps, but then diverged to FID > 260. The corresponding samples do not resemble realistic images.

---

### Decision · Program_Chairs · 2021-01-07
**Final Decision**

**Decision:**

Reject

**Comment:**

The paper receives a mixed rating, with R3 rates the paper above the bar, R1 and R2 rates marginally above the bar, and R4 recommends rejection. The cited positive points include 1) decomposing image generation into first synthesizing segmentation masks and then converting segmentation masks to images, and 2) good results comparing to Progressive GAN and BigGAN. R4 raises several concerns, including the novelty concern and unconvincing experimental validation. After analyzing the papers, the reviews, and the rebuttal, the AC finds the arguments made by R4 more convincing. Decomposing image generation to a two-step approach has been illustrated in the prior work [Wang & Gupta ECCV 2016, Hong, Yang, Choi, Lee CVPR 2018]. The proposed method does not provide additional insights. The provided experimental results are not very convincing, either. As the proposed setting assuming the availability of segmentation masks, it is not surprising that it outperforms the unconditional baselines. Overall, the AC believes the paper does not have enough novelty to justify its acceptance and would recommend rejection of the paper.